# A Rare Case of Tularemia Complicated by Rhabdomyolysis with a Successful Outcome

**DOI:** 10.3390/medicina57050449

**Published:** 2021-05-05

**Authors:** Ieva Kubiliute, Birute Zablockiene, Rasute Paulauskiene, Giedrius Navickas, Ligita Jancoriene

**Affiliations:** 1Center of Infectious Diseases, Vilnius University Hospital Santaros Klinikos, 08410 Vilnius, Lithuania; birute.zablockiene@santa.lt (B.Z.); rasute.paulauskiene@santa.lt (R.P.); ligita.jancoriene@santa.lt (L.J.); 2Clinic of Infectious Diseases and Dermatovenerology, Institute of Clinical Medicine, Faculty of Medicine, Vilnius University, 01513 Vilnius, Lithuania; 3Center of Cardiology and Angiology, Cardiac Intensive Care Department, Vilnius University Hospital Santaros Klinikos, 08410 Vilnius, Lithuania; giedrius.navickas@santa.lt

**Keywords:** tularemia, ulceroglandular tularemia, *Francisella tularensis*, rhabdomyolysis

## Abstract

We present a case of tularemia complicated by rhabdomyolysis in a 43-year-old male who presented with fever, swelling, and pain of the right groin and a history of a week-old tick bite. Empirical parenteral amoxicillin/clavulanic acid treatment was initiated. Suspecting tularemia, parenteral gentamycin was added. Later, the patient started to complain of muscle pain, weakness, and difficulties in breathing and walking. Heightened levels of creatine kinase and myoglobin concentration (42,670 IU/L and >12,000 μg/L, respectively) were found. Due to rhabdomyolysis, large amounts of intravenous fluid therapy were initiated to prevent kidney damage, continuing intravenous antibiotic therapy. *Francisella tularensis* IgG in serum was found to be positive only on the sixteenth day of hospitalization. Upon discharge, the laboratory analyses returned to normal levels, and the patient was in good condition. The successful outcome could be associated with the early appropriate therapy of tularemia and its rare complication of rhabdomyolysis.

## 1. Introduction

Tularemia is a rare, contagious, vector-borne, and zoonotic infectious disease caused by the bacterium *Francisella tularensis*, occurring mostly in the Northern Hemisphere [1,2,3,4,5]. The confirmed tularemia prevalence in European Union and European Economic Area (EU/EEA) countries was 0.3 cases per 100,000 people in 2019 [6]. Tularemia has six major clinical forms [2,3,5,7,8]. The most common form, ulceroglandular tularemia, is usually the result of direct skin inoculation by an infected tick or deerfly or the handling of infected animals. It is characterized by skin lesions and localized lymph node swelling, pain, and the development of ulceration and suppuration if left untreated [2,3,7,8]. Nonspecific symptoms, including fever, fatigue, malaise, chills, and headache, are often present. The severity of the disease depends on the entry route of *Francisella tularensis*, the virulence of the bacteria strain, and the condition of the host’s immune system [2,3,7,9]. Severe tularemia cases may be complicated by pneumonia, acute renal injury, intravascular coagulopathy, meningitis, pericarditis, peritonitis, osteomyelitis, sepsis, and septic shock [9,10,11,12]. Here, we present a rare case of ulceroglandular tularemia complicated by rhabdomyolysis with a successful outcome.

## 2. Case Report

A 43-year-old Lithuanian male was admitted to a hospital because of fatigue, malaise, fever up to 39 °C, and swelling and pain of the right groin. These symptoms lasted for three days. One week prior, the patient removed a tick from the medial side of the right knee. The patient was previously healthy and had no chronic diseases. He reported keeping a dog and a cat. Although he owned a sheep farm, he did not have direct contact with these animals.

Upon admission, his vital signs were: temperature 36.8 °C, blood pressure 100/60 mmHg, and pulse rate 84 beats/minute. An ulcer 1 cm in diameter on the medial side of the right knee was observed, where the tick had been attached. There was a palpable and painful 3 cm × 6 cm lymph node conglomerate in the right groin area, with erythema of the overlying skin. A laboratory analysis revealed a normal white blood cell count (6860 cells/μL) and increased concentration of C-reactive protein (CRP) (118.8 mg/L). The urine analysis showed no essential changes. The chest X-ray was normal, and negligible hepatosplenomegaly was detected during an abdominal ultrasound. Two blood cultures and culture from the ulcer yielded negative results. Empirical parenteral amoxicillin/clavulanic acid (1.2 g four times a day) was initiated. Suspecting tularemia, parenteral gentamycin (80 mg three times a day) was added, although the initial serological test, Western blot (WB) to IgG antibodies of *Francisella tularensis*, was negative.

After three days of treatment, the pain and size of the lymph node conglomerate diminished, erythema of the overlying skin disappeared, and the concentration of CRP decreased to 65.3 mg/L, but the febrile fever persisted. The patient started to complain of myalgia, heart palpitations, and breathing difficulties on the fourth day of hospitalization. It was difficult for the patient to walk and even to open his mouth due to myalgia. Repeated laboratory analysis revealed leukocytosis in serum (15,110 cells/μL), elevated concentrations of CRP—184 mg/L, alanine aminotransferase (ALT)—609 IU/L, aspartate aminotransferase (AST)—1662 IU/L, creatine kinase (CK)—42,670 IU/L, lactate dehydrogenase (LDH)—1446 IU/L, troponin I—2201 ng/L, myoglobin—>12,000 μg/L, and potassium—6.3 mmol/L. The concentration of creatinine remained normal. The urine sample was dark brown, and proteinuria of 1.0 g/L and 8–9 erythrocytes per visual field were observed. Rhabdomyolysis was suspected, and the patient was transferred to the intensive care unit. A large amount of intravenous fluid therapy was initiated to prevent kidney damage, continuing intravenous antibiotic therapy. After two days, the patient’s muscle pain decreased, and there were no signs of acute kidney damage or fever. He was transferred to the Department of Internal Diseases, continuing the intravenous antibiotic therapy and rehydration. 

The patient responded well to treatment: the patient’s general condition improved, his body temperature decreased to a normal level, the muscle pain disappeared, the lymph nodes in the right groin area decreased in size, and the wound on the medial side of the right knee healed. CRP, ALT, AST, CK, troponin I, myoglobin, LDH, and potassium concentration decreased to normal levels. On the seventeenth day of hospitalization, the patient was discharged from the hospital in a satisfactory condition. 

Analyses for various infectious agents were performed. Serological tests for tick-borne encephalitis virus, *Borrelia burgdorferi*, Cytomegalovirus, viral hepatitis B and C, Epstein–Barr, HIV infection, and syphilis, as well as molecular tests for influenza A and B and respiratory syncytial virus infection, were negative. During hospitalization, three WB tests for detecting *Francisella tularensis* IgG antibodies were conducted: the first one was negative (on the first day of hospitalization), the second one was faintly positive (threshold) (on the eighth day of hospitalization), and only the third one was positive (on the sixteenth day of hospitalization). Based on this presentation, the patient was diagnosed with ulceroglandular tularemia complicated by rhabdomyolysis.

## 3. Discussion

Tularemia may be a diagnostic challenge for physicians, especially in countries where this disease is rare. The confirmed tularemia prevalence in the EU/EEA ranged from 0.1 to 0.3 cases per 100,000 people in 2015–2019 [6]. It is mostly diagnosed using serological tests that determine antibodies against *F. tularensis*, but these antibodies can be detected only 10–20 days after the onset of the infection [3,9]. We probed IgG antibodies using WB, the only method currently available in Lithuania. The initial WB against tularemia was negative, as it should be in the beginning of the disease. Even though it could confuse the physician, he/she might suspect tularemia according to characteristic clinical signs, an ulcer in the site of the tick bite and regional lymphadenopathy, and initiate the appropriate treatment in a timely manner. 

We treated the patient with parenteral amoxicillin/clavulanic acid and gentamycin from the first days of hospitalization. The reason for adding gentamycin was the clinical suspicion of tularemia. It is known that *F. tularensis* is resistant to beta–lactam antibiotics [5]. Parental aminoglycosides are recommended as the first line of treatment for severe tularemia [3,5,7]. The early initiation of gentamycin can influence the successful outcome, even in cases of rhabdomyolysis.

Rhabdomyolysis with tularemia was first reported in 1985, when Kaiser et al. reported four cases. We found only 11 described cases in English language publications in PubMed [10,11,13], and our patient could be the twelfth case of tularemia complicated by rhabdomyolysis. Ten men (including our case) and only two women were affected, most of them developing acute kidney injury [10,11,13]. Although tularemia affects both males and females, the number of cases is higher in males. The male-to-female ratio in the EU/EEA was 1.5:1 in 2019 [6]. This tendency can be related to greater occupational exposure and recreational outdoor activities in men, like hunting and fishing [2]. Out of the 12 found and analyzed tularemia complicated by rhabdomyolysis cases, there were four (33.3%) lethal outcomes reported. Although there is not enough detailed data on the analyzed cases, two (50%) of the cases did not get the appropriate antibacterial treatment—it was started late or did not include aminoglycosides. Worse prognosis might be related with the level of CK (the patients who died had CK levels higher than 23,000 IU/L) and the patient age (75% were older than 60 years) [10,11,13]. The good outcome of our patient could be associated with the early recognition and appropriate treatment of the disease.

## 4. Conclusions

In conclusion, tularemia should be suspected in a case of lymphadenopathy following the tick bite, even if the initial serological tests are negative. Rhabdomyolysis is rare but treatable complication of tularemia, and it is important to start an appropriate therapy early to avoid a threat to life.

## Data Availability

The data presented in this case report are available on request from the corresponding author.

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
