# Peer review of "A Rare Case of Tularemia Complicated by Rhabdomyolysis with a Successful Outcome"

_medicina, 2021, doi:10.3390/medicina57050449_

Round 1

Reviewer 1 Report

The authors describe a severe case of tularemia with rhabdomyolysis. The manuscript describes the case and therapeutic measures in clear language.

I have only a few minor comments:

Section Case Report :

line 63,65 and 66: Please add thousands separator.

line 83: "turned out" instead of "came back"

line 83: "conducted" instead of "submitted"

line 83: "was" instead of "came back"

Section Discussion:

Lines 90/91: Please add numbers on tularemia in humans and animals in the last years. 

line 108: What is the sex ratio usually in tularemia cases. Please add and discuss. 

line 111: "did not include" instead of "not included"

Author Response

Response:

Section Case Report:

Lines 63, 65 and 66: thousands separator was added.

Line 83: “came back” was changed to “turned out”.

Line 84: “submitted” was changed to “conducted”.

Line 86: “came back” was changed to “was”.

Section Discussion:

Lines 91/92 (former lines 90/91): the changes of tularemia prevalence in Europe in recent years presented by European Centre for Disease Prevention and Control was added. The tularemia prevalence in animal population is not exactly known therefore this information was not included in the manuscript.

Lines 115-119 (former line 108): the sex ratio and probable reasons of this tendency in tularemia cases were added.

Line 121 (former line 111): “not included” was changed to “did not include”.

Reviewer 2 Report

Great case report describing the 12th reported case of tularemia complicated by rhabdomyolysis. The case report was succinct and to the point, yet covered background information. Good conclusions encouraging the consideration of tularemia despite a negative serological test early on. 

I would insert "the bacterium" into lines 25-26 to read as follows< "...zoonotic infectious diseased caused by the bacterium Francisella tularensis,..."

More details on where the patient is from, besides Europe, would be beneficial to the reader. 

Besides that, I have no further comments. Nice manuscript.

Author Response

Response:

Lines 25-26: “the bacterium” was inserted.

Line 41: the nationality of the patient was included.
